# Effective Patient–Dentist Communication with a Simulation System for Orthodontics

**DOI:** 10.3390/healthcare11101433

**Published:** 2023-05-15

**Authors:** Yi-Cheng Chiang, Fan Wu, Shu-Han Ko

**Affiliations:** 1Department of Information Management, National Chung-Cheng University, Chiayi 621301, Taiwan; sofia@mis.ccu.edu.tw; 2Taichung Tzu-Chi Hospital, The Buddhist Tzu Chi Medical Foundation, Taichung 427213, Taiwan; 3Institute of Healthcare Management of Information System, National Chung-Cheng University, Chiayi 621301, Taiwan; tommy@mis.ccu.edu.tw

**Keywords:** patient communication, simulation, orthodontics, dentistry

## Abstract

Orthodontic treatment is a common dental treatment. A complete treatment often takes several years and is costly. In order to increase the degree of satisfaction and reduce the gap between the expectations of the patient and the limitations of orthodontics, orthodontists and patients should have sufficient communication. A simulation system can provide a good channel for communication between orthodontists and patients. This paper proposes a 3D dentist–patient communication system for the simulation of orthodontics in advance. The system collects the clinical paradigms of orthodontic cases, which must contain photos from before, during, and after maxillofacial treatment. This research simulates treatment processes by tuning a 3D virtual model of the oral and maxillofacial areas, including the face, mouth, and teeth, to demonstrate the processes of clinical paradigms. These 3D models could be edited and synthesized to generate new 3D models for simulation with the 3DS Max tool. In addition, the proposed system provides a function for the orthodontist to manually operate the 3D virtual model, such as tooth arrangement, morphing of the gums with movement of the teeth, the placement (attachment) of arch-wires and brackets, and changes of facial appearance. The orthodontist can demonstrate the treatment processes, show movements of the teeth, and answer possible questions from the patient about the treatment processes by using the 3D model. To show the effectiveness of the proposed system, a questionnaire about the system was also used to demonstrate its acceptance, usability, and validity. Qualitative interviews with dentists and questionnaires for patients about the system are both performed. The results showed that the proposed system is an effective vehicle for communication between patients and orthodontists.

## 1. Introduction

Because of improvements in quality of life, people have become aware of aesthetics in daily life and their appearance. Today, medical treatments not only cure physical diseases, but also satisfy the psychological requirements that people have for their mind and appearance. Orthodontic treatment is a common way to cure malocclusion, which may cause dental caries and periodontitis; in addition, orthodontic treatment is a medical treatment that can improve dental aesthetics, beautify the facial appearance to build confidence and develop good relationships, and remedy pronunciation and chewing barriers [1,2].

Orthodontic treatment is complicated, as it takes place over a long duration, and is often expensive. In addition, the results of the treatment sometimes cannot fulfill the expectations of the patient [3]. Thus, medical disputes often arise because of the asymmetry of information between patients and orthodontists. Some studies have shown that good communication between the orthodontist and the patient has significant effects on the success of orthodontic treatment [4,5,6]. In other fields, patient–physician communication and relationships were shown to be critical factors in the treatment of many diseases related to patients’ quality of life, satisfaction, self-confidence, and self-management in medical behavior [7,8,9].

Traditionally, in orthodontics, many real devices, such as tooth models and casts, have been used to show the process of orthodontic treatments [10]. Due to advancements in computer technology, many studies have incorporated computer simulations as a vehicle for communication in orthodontics [11,12]. A simulation can be used to support decision making before a treatment or surgery, predict the results of treatment in advance, and present the procedure of the treatment [13]. Nowadays, advanced computer technology can be used with three-dimensional (3D) graphics to generate 3D virtual dental models to replace real devices and objects, thus reducing the cost and manual errors and even saving storage space. To provide an educational function, an orthodontics computer-assisted instruction system was developed by Wang et al. [12]. This CAI system collected many past orthodontic cases and constructed a 3D animated video for each case. The system could immediately find the most similar past cases through the technique of case-based reasoning. When an orthodontist needed to explain a treatment to patients, the system could provide the most similar case and show it with a 3D animation, thus serving as a vehicle for communication between the patient and orthodontist prior to the treatment. However, the treatments were based on past orthodontic cases; they did not consider the status of current patients. In addition, the system did not provide simulations of tooth arrangement for patients. Until now, there have been no orthodontic simulation systems with the capability of providing just-in-time 3D simulations of processes from pre-treatment to post-treatment and all potential operations on each tooth, such as movement, rotation, or extraction, under the consideration of the current patient’s status.

In this study, we developed a just-in-time 3D simulation system for orthodontics. It provides simulation functions such as tooth movement, rotation, extraction, arrangement, and facial morphing from pre-treatment to post-treatment, all of which are operated in an intuitive 3D way, such as by picking, dragging, and rotating the 3D model with a mouse. By using these intuitive 3D operations, the orthodontist can arrange the 3D model in real time according to the status of the patient. The 3D simulation and related treatments of the current patient can be selected as a new paradigm for teaching and communication materials for later patients or interns. Orthodontists and dentists can use this system to present the dynamic processes of orthodontic treatments to patients. As a result, patients can better understand their treatment procedures, and their satisfaction can be increased.

To show the effectiveness of the system, we first performed qualitative interviews with experienced orthodontists. First, the results of the qualitative interviews showed that the orthodontists all agreed that the system could be an effective and efficient tool for education and communication with patients. Secondly, we also validated the effectiveness of the system. After assessing patients that needed orthodontic treatment with a questionnaire, the results showed that the rate of patients’ intentions to receive treatment increased by 22% when the system was used to communicate with patients, rather than using the traditional way of communication. With dentists and patients giving positive recognition, this system reaches the significant goal of educating orthodontists and patients, it is an effective tool for communication, it reduces the gap in expectations between orthodontists and patients, and it increases the satisfaction of patients.

## 2. Materials and Methods

### 2.1. Materials

In order to present the simulation results, we used basic 3D models consisting of the skull, mandible, maxilla, teeth, masseters, soft tissues, etc. These 3D models could be edited and synthesized to generate new 3D models for simulation with the 3ds Max tool, which is a professional 3D computer graphics program for making 3D animations, models, and images and is developed and produced by Autodesk Media and Entertainment. This 3D tool is frequently used by video game developers, many commercial TV studios, and architectural visualization studios. It is also used for movie effects and movie pre-visualization, and it has modeling capabilities and a flexible plugin architecture that is run on the Microsoft Windows platform with its own scripting language. To generate the new model, we first removed the unnecessary basic 3D models, such as those of the skull and masseters, which the system did not use in the simulations of orthodontic treatments. In the next step, we separated the 3D tooth models, which were initially connected with each other, into a set of 32 disconnected tooth models such that these 32 models could be individually operated upon in the system. In addition, we designated a pivot for each model of teeth at its center as the mass point for later operations, such as rotating, pulling, and pushing the teeth.

Figure 1 shows the movement of the mass point of a tooth model, where we designate the pivot of each model of tooth to its center from the outside of the tooth (seen in Figure 1a) to the center of the tooth (seen in Figure 1b). Firsly, the mass point is the origin with three rays along *xyz* axis, in red green and blue, emitted from it. With the designation of mass point, the later operations, such as rotating, pulling, pushing, etc., can be performed near the reality. To simulate the process of the orthodontic treatment of a patient, we also created 3D arch-wire and bracket models that could be attached to the basic 3D models. The original soft tissues were also separated into the lower half of the face and the gums of the teeth. By using an edit poly-modifier that operated on the lower half of the face in the 3D model, we created 10 different morph results in advance for the appearance of the face, i.e., open mouth, open lips, plumping/sinking of the cheeks, plumping/sinking of the cheek on the left side, plumping/sinking of the cheek on the right side, rising/descending mouth, protrusion/recession of the mouth, protrusion/recession of the upper lip, protrusion/recession of the lower lip, and protrusion/recession of the jaw, as seen in Figure 2a–j. The edit poly-modifier was provided by the 3ds Max tool, which also provided other functions, such as those for creating new polygons, editing the properties of polygons, changing the position and color of each vertex, and capping the borders, i.e., patching a surface to cap the hollow part surrounded by a plane in the 3D models. In addition, the gums were separated into the upper and lower gums via the edit poly-modifier.

During animation rendering in cinema post-processing, computer-generated imagery, and video games, shaders are widely used to produce a range of effects. This study attached complex shaders to the materials and textures of these basic 3D models to provide finer rendering results under the consideration of light reflection, antialiasing, and so on, the position and color (hue, saturation, brightness, and contrast) of all pixels, vertices, and/or textures were used to construct the final rendered image. Note that the more complex shaders needed more powerful graphic hardware for rendering. To increase the rendering speed and performance, we could simplify the properties of the materials, such as by using a diffuse color, which was the color reflected from the 3D model when it was illuminated by daylight or artificial light, and a bump map, which built a visual effect in such a way that the lighter (whiter) areas of the map appeared to be raised and the darker (blacker) areas appeared to have sunk. In the final step, we attached a smooth modifier to these 3D models to make the edges of the polygons in the models sleek. Figure 3a shows a 3D model of the rendered results for the face, while Figure 3b shows a 3D model of the rendered results for the teeth and gums after the editing of basic 3D models described above.

We used many polygons to emulate each 3D model. Clearly, the larger the numbers of polygons and vertices was, the higher the resolution of the model would be. However, the lag when executing the simulation increased if the number of polygons increased. Table 1 shows the numbers of polygons and vertices used in the 3D models in this study to make a tradeoff considering the resolution of the models and the lag time of the simulation.

### 2.2. The 3D Simulation Functions

In orthodontic treatments, a dentist examines the characteristics of a patient, makes a diagnosis, and determines the patient’s symptoms. To simulate the process of orthodontic treatment, the proposed system first needed to emulate all possible symptoms of patients. Some convenient simulation functions were provided to emulate the symptoms, such as scaling, moving, and rotating the teeth. In addition, after activating the extraction (or implantation) option, a tooth could be easily extracted (or implanted) by clicking on the tooth (or the position of the missing tooth). According to the orthodontic knowledge provided by previous work [14,15,16], this system provided six types of tooth movements (i.e., displacements) to describe the force used during orthodontic treatments, namely, tipping, translation, rotation, intrusion, extrusion, and torque. Figure 4 shows the six types of movement of a tooth with force simulated in this system; the grey circle shows the anchor of the tooth, the white arrow *F* shows the force and its direction, and the red arrow indicates the direction of movement of the tooth. Tipping in Figure 4a is the simplest tooth movement when the force pulls or pushes upon the tip of the tooth, resulting in the crown of the tooth moving much more than the root. Translation movement in Figure 4b, called bodily movement, describes the crown and root being moved in the same direction at the same time. Rotation in Figure 4c means the tooth being moved around its long axis (rotating axis). Intrusion in Figure 4d is a movement of the tooth in the apical direction with very light forces. Opposite to intrusion, extrusion in Figure 4e is the movement of the tooth in the occlusal direction. Finally, torque in Figure 4f means the root being moved under the minimal movement of the crown.

Along with the movement of the tooth, the proposed system simulated the associated changes in the gums caused by the movement. That is, the related gums were morphed and adapted to the current status of the arrangement of the teeth. In addition, the system provided the function of emulating the treatment of skeletal malocclusion. Users could simulate such treatments by dragging all gums and teeth in the either upper or lower parts of the mouth. The related gums could morph accordingly; that is, the system could cap the hollow area when a tooth was extracted or shrink the gums to the appropriate position when a tooth was implanted.

Through the combination of the above potential arrangements of the teeth and gums, the system was able to present all of the possible symptoms of patients. The next task was to emulate the process of treatment. First, users could add the arch-wires and brackets to the teeth according to their expectations. To simulate treatments with arch-wires and brackets, the system provided the components of brackets that could be attached to the teeth to be corrected. After adding the brackets to the related teeth, arch-wires could then be added to connect every two adjacent brackets. Note that in real life, an arch-wire is a rigid metal wire, and the curve between two brackets is not always a straight line. To model the curve, the system added a point called a controlling node on the arch-wire between two adjacent brackets, which allowed the user to tune the bending degree of the arch-wire between these two brackets. With the tuning of the controlling nodes (i.e., moving the position or rotating the angle of orientation), the simulation could vividly model the curves of the rigid metal lines of the arch-wires and prevent the arch-wires from penetrating the gums and teeth.

Figure 5a shows a situation in which a straight arch-wire penetrated a tooth. After tuning the controlling node, the arch-wire can be bended as a curve along the contour of the teeth. Figure 5b shows the final arch-wire after hiding the controlling node, making it look like a real situation. It is worth noting that the system used a curve to simulate the arch-wire between two brackets, and the curve surrounded the contours of the corresponding teeth as closely as possible. In most cases, the curve of the arch-wire could fit the contours formed by the two surfaces of the teeth.

Next, we added a 3D model of the lower half of a face to present the appearance of the face of a patient. The system provided the following functions: opening of the mouth, opening of the lips, plumping/sinking of the cheek, plumping/sinking of the cheek on the left side, plumping/sinking of the cheek on the right side, rising/descending of the mouth, protrusion/recession of the mouth, protrusion/recession of the upper lip, protrusion/recession of the lower lip, and protrusion/recession of the jaw, as shown in Figure 2. For each function of morphing, the system provided a sliding bar that could be used to tune the morphing degree from zero to one, where a degree of zero meant that the facial feature was the same as that in a normal face, while a degree of one meant that the facial feature underwent the maximum amount of morphing. Based on the 10 morphing functions, the user could individually use or arbitrarily combine these morphing functions to simulate the facial appearance of patients.

With these four simulation functions (i.e., tooth arrangement, morphing of the gums with movement of the teeth, the placement (attachment) of arch-wires and brackets, and changes of facial appearance), the user could manually arrange the positions of the teeth and the facial appearance according to the patient’s status during pre-treatment. A 3D model mimicking the status during pre-treatment could be set as the starting point for an orthodontic treatment. Dentists can arrange the positions of a patient’s teeth again according to the expected or default post-treatment status as the end point of the treatment. After that, the system can then automatically perform an orthodontic treatment; that is, the arrangement of the teeth can smoothly evolve (or change) from the starting point to the end point of the treatment. The following operations were adopted in the system to allow the evolution of the arrangement of the teeth to be realized. The first was a moving function that could move a 3D model (in this study, this refers to a tooth) from a certain position to a designated position; the second was an orientation angle function that could rotate the angles of the 3D model along the *x*, *y*, and/or *z* axes from an initial orientation angle to a target angle. The system provides the rotation of the tooth model, shown in Figure 6. Figure 6a shows the tooth model observed from the *xy* plane, while Figure 6b shows the model being rotated and observed from the *yz* plane. Generally, these two operations are sufficient for mimicking the effects of orthodontic treatments on teeth. As for the morphing of the facial appearance caused by the treatment, the system provided a third operation, i.e., a multi-morphing function, which operated on two 3D models (i.e., models for the starting point and end point) that needed to have the same number of vertices, regardless of if they looked different. Note that in this study, the pre- and post-treatment facial appearances of a patient were combinations of the results of the 10 types of morphing of the face, each of which addressed an independent facial feature to constitute the facial appearance of the patient. The tuning of the 3D model of the lower half of the face by dentists was to set the individual degrees of morphing of these 10 morphing function to emulate both the appearance of the patient before the treatment and the expected appearance after the treatment. The provided multi-morphing function first took the degree of morphing of each facial feature (i.e., base morphing) at the start and end of the treatment. Then, the function gradually morphed the facial appearance from the starting point to the end point, thus demonstrating the morphing of the facial appearance caused by the treatment.

According to clinical experience, the progress of an orthodontic treatment can be divided into several phases (or milestones). Each milestone is a tooth arrangement with only small differences (including extracting a tooth, pulling some of the teeth, implanting a tooth, etc.) from the tooth arrangement in the previous milestone. The system allows dentists to divide treatments into several phases and arrange the teeth as they are expected to be in each phase (or milestone). The three operations described above (i.e., the moving, orientation angle, and multi-morphing functions) can be used to simulate the progress of an orthodontic treatment from one phase to the next. If the simulated progress is not vivid enough, the dentist can add another treatment status during the treatment according to their expectations.

Note that regardless of if the dentist uses the default post-treatment status or designates their expected post-treatment status, they need to designate two teeth on the right and left sides of an arch and the anchors to which they will both be attached by a band that is used to pull and align the teeth in front of them. In general, the two anchoring teeth are in a symmetric pair in an arch. However, the symmetric pair may not always be in the same horizontal plane. Even worse, it might be that no symmetric pairs can be used for the treatment because of the extraction of a tooth; that is, the two anchoring teeth might not always be a symmetric pair. Clearly, this flexibility provides the system with the ability to model all situations of malocclusion in a patient. To simulate a treatment with two asymmetric anchoring teeth or a symmetric pair that has a deviation away from the default position, the system can automatically calculate the expected positions on the *x* and *y* axes (i.e., the horizontal plane) of each tooth in front of the anchoring teeth and then calculate the expected position on the *z* axis (i.e., the vertical axis) of each of these teeth according to their length. In each time unit, if the teeth are moved from the starting point to the end point, the system uses the operations for movement and setting the orientation angle to move the teeth step-by-step from the starting point to the end point of the treatment. The arch-wires attached to the teeth will be moved accordingly, making it look like the movement of the 3D model of the teeth is achieved by pulling the arch-wires. The progress of the whole treatment is shown, and the patient and dentist can pause and move forward and backward for discussion, or even zoom in, zoom out, and change the viewpoint.

For the movement of the teeth, the system recorded all information about the movement in each time unit, including the position of each tooth, bracket, and controlling node of the arch-wires. According to the progress of tooth movement, the facial profile was also morphed. The system also calculated the degree of morphing of the face of the patient in terms of the 10 types of base morphing in each time unit, and the system then used the multi-morphing operation on the lower half of the face to morph it step-by-step from the starting point to the end point of the treatment.

To implement the above simulation functions and operations, we developed the patient–dentist communication system via Virtools, which is a tool kit provided by the Virtools^TM^ company that aims to support the development of 3D interaction systems and provides many basic methods for 3D modeling. For example, Virtools provides functions for morphing the appearance of a 3D model, translating the model’s position or rotating its orientation angle, and controlling its visibility. The above functions were chained in a sequence to model the progress of treatments.

This study first referred to an orthodontic textbook [17] to survey clinical procedures from the beginning to the end. Knowledge and clinical guidelines were collected to provide past paradigms for dentists such that they could immediately present the treatment paradigms. Therefore, dentists could conveniently communicate with patients about paradigm cases.

To emulate the treatments received by current patients, we first collected over 900 records of orthodontic cases from some dental clinics in Taiwan. For the quality of the simulations, we filtered the cases according to the following criteria: the cases needed to have (1) clear facial photographs in the frontal view and profile view, (2) sufficient intra-oral photographs in the frontal, right, left, upper, and/or lower occlusal view, (3) panoramic and lateral cephalometric radiographs, and (4) complete orthodontic treatment histories. The cases were filtered and categorized into class I, class II, and class III according to the Angle classification [18]. Our research only focused on Angle class II, since the cases of class II malocclusion occur most frequently [19]. According to previous studies [19,20,21], the symptoms of Angle class II can be divided into fifteen types, i.e., crowding, skeletal overjet, crossbite, overbite, high canine, forced eruption, midline deviation, prosthodontic needs, protrusion, open bite, diastema, spacing, rotation, lost teeth, etc. The collected cases were classified according to the fifteen symptoms and stored in the database. Experts comprehensively validated each collected case; these cases were used to generate the evolution of the treatment simulations, and they were then adopted in the system.

In sum, with the proposed system, dentists can communicate with patients in two ways. The first way is by demonstrating the process of treatment for the current patient. Figure 7 and Figure 8 show actual photographs and the corresponding 3D models of a simulation of the arrangement of the face and teeth before treatment and during treatment. The dentist can arrange the arrangement of the face and teeth according to the patient’s pre-treatment status. The system can then copy these 3D models in the pre-treatment as the prototype for the dentist to arrange for the models during the treatment under the consideration of the malocclusal status for each tooth and the limitations (e.g., constraints due to gum or periodontal disease, tooth decay, and extraction number) of orthodontics. After the dentist chooses the pair of anchoring teeth on which to attach the arch-wire to pull the frontal teeth, the system can then automatically arrange the positions of the teeth step-by-step from the starting point to the end point. More in-treatment statuses can be input during the evolution of the simulation to fit the progress of the treatment in each phase. Dentists and orthodontists can display the full process from the pre-treatment status, through several in-treatment statuses, to the post-treatment status. When displaying this, the dentist can communicate with the patient and explain the related procedures. The second way is to display past paradigmatic cases. Dentists and orthodontists first diagnose the symptoms of the current patient and select the most similar past paradigmatic cases. Since the simulations of the paradigmatic cases are prepared beforehand, there is no wait for the patients and dentists to watch their treatments. Clearly, the more paradigmatic cases the system has, the more similar the treatment processes shown to the patient will be.

## 3. Results

After constructing the communication system, we performed two empirical studies to show its effectiveness. The first study was used to examine whether the system met the users’ requirements and achieved the desired goals. The second was used to evaluate and compare the communication capabilities of the system with traditional communication from the point of view of patients.

### 3.1. Qualitative Interviews with Dentists about the System

We first interviewed five orthodontists, all of whom had more than two years of practice. These five orthodontists consisted of three males and two females, and they are denoted as A, B, C, D, and E. In order to assess whether the simulation functions and displays of the system met their needs in their clinical practices, the main focus of the interviews was the satisfaction with the simulation functions provided by the proposed system in comparison with traditional physician–patient communication. The overall results of the interviews were positive, and they are summarized in the following.

All orthodontists agreed that through the system, communication with patients was clearer than the traditional means of communication; all orthodontists believed that they could vividly explain the orthodontic treatments with the help of the system, especially since the system could express the evolution of a treatment step-by-step from before and during to after. They also agreed that these simulation functions could increase the intentions of patients to receive orthodontic treatments and could improve the quality of communication between orthodontists and patients. All orthodontists agreed that most of the symptoms of Angle class II could be found in the cases collected in the system; however, they hoped that the number of collected cases would be increased to cover all possible symptoms and combinations thereof.

Four of the five orthodontists appreciated that the system provided intuitive ways of operating the 3D model to emulate all possible symptoms, such as by clicking and dragging the mouse. In addition, three of the five orthodontists believed that the simulation functions provided in the system could help patients in decision making before a clinical treatment and make the patients aware of the potential limitations (i.e., constraints due to gum or periodontal disease, tooth decay, and extraction number) of treatments.

From these results of the interviews, we could see that the proposed system provided vivid simulation functions for helping educate patients on the procedures of orthodontic treatments and their potential limitations in advance. Clearly, the system is an effective tool for communication between dentists and patients, and it provides dentists with more capabilities for expression.

### 3.2. Questionnaires for Patients about the System

This study also evaluated the communication capabilities of the proposed system through a questionnaire. The population of subjects consisted of 36 patients, which included 15 males and 21 females, whose individual ages were from 12 to 27. The questionnaire consisted of eight statements (the details can be seen in Appendix A) focused on four aspects, namely, convincing ability, communication quality, degree of understanding, and rate of change in intentions (i.e., to accept a treatment). The first two aspects evaluated the system’s usability and acceptance; statements 1, 2, 3, and 4 in the questionnaire were designed for this purpose. The last two aspects evaluated the system’s understandability and effectiveness; statements 5, 6, 7, and 8 were designed for this purpose.

Likert’s five-point scale was adopted for scoring in the questionnaire; that is, the patients replied to each question by ticking a score from 1 to 5, indicating “strongly disagree” to “strongly agree”. For dentist–patient communication about orthodontics, this study added an extra phrase that was called phase two, in addition to the traditional means of communication, which was called phase one. In detail, in phase one, the dentists communicated with patients by using the traditional means of communication with the photos and casts of past cases, as well as explanations. The dentists explained the notices (including the costs and duration of the treatment), the procedures of the treatment, the results of similar past cases after treatment, and the estimated results for the current patient. After the communication in phase one, the patients are asked to fill out a questionnaire to evaluate the effectiveness of the traditional means of communication. After that, the dentist started with the communication in phase two; that is, the dentist communicated with the patients by using the proposed system. The communication involved the display of past paradigmatic cases and just-in-time simulations of the current patient with the proposed system. That is, if the symptoms of the current patient were similar to those in past paradigmatic cases that were in the system, the dentist could immediately display the past case and communicate with the patient. However, if the symptoms of the current patient were dissimilar to those in the past paradigmatic cases or the system did not have enough past paradigmatic cases for the current case, the dentist or dental hygienist could arrange and tune 3D models in the system to emulate the status of the current patient before treatment within about 5 min. According to their experiences, they then arranged these 3D models to show the status during and after the treatment. Finally, they activated the system to display the treatment step-by-step from before and during to after. After the two phases, the patients were asked to fill out the questionnaires to evaluate the effectiveness of the additional communication.

By analyzing the two questionnaires for the two phases, we obtained the scores of patients who evaluated the system with respect to the aspects of convincing ability, communication quality, degree of understanding, and rate of change in intentions. Figure 9 shows the average scores for these four aspects. We could see that with the additional help of the proposed system, the communication capabilities were greater than those of only traditional communication in all four aspects. In the aspect of convincing ability, the communication with the proposed system obviously outperformed traditional communication. This was because some patients, especially adolescents, were asked to receive orthodontic treatments by their parents and elders, rather than of their own will. For such patients, there was some hesitation or even refusal to receive the treatment. After viewing the 3D display of the proposed system, which emulated the appearance of the patients from before to after the treatment, they knew that the treatment could really improve their appearance. From the vivid display, they were convinced or had a higher incentive to receive the treatment. For the aspect of communication quality, although three more minutes were needed, on average, for the additional communication with the proposed system in phase two, the entire communication process (i.e., phase one and phase two) allowed the patients to perceive more details of the evolution of their orthodontic treatments. From the results for statements 3 and 4 of the questionnaire, we could see that the patients considered that the extra time taken to explain the treatments with the proposed system was acceptable and that they could better understand the process of the treatments with the communication provided by the system. Thus, we concluded that these patients received high-quality communication through the system. For the aspect of the degree of understanding, the traditional means of communication inherently could not dynamically display the details of the process of an orthodontic treatment, such as the displacement of teeth with/without the extraction of a tooth (or teeth) and the morphing of the face. The dentist could only give an abstract explanation that lacked concrete images of the evolution of the treatment. According to the results for statements 5 and 6, patients agreed that they can better understand the treatments with the orthodontic evolutions displayed by the system. For the aspect of rate of change in intentions, which was the most practical concern of dentists, we could see that the rate of intending to receive orthodontic treatment when explaining in the traditional way was lower than that when explaining with the proposed system. It was hypothesized that this was because some patients did not have full confidence or hesitated to receive a treatment when facing unclear and uncertain changes in the appearance of their teeth. The proposed system could display the 3D models of their post-treatment appearance, resulting in the fact that the rate of intending to receive the treatment increased if it was explained with the proposed system.

Given that the communication with the proposed system outperformed the traditional means of communication in the above four aspects, we were curious about the characteristics of the population profile and the advantages of the proposed system that were significant. Next, we adopted a statistical theory to test whether the improved performance of the proposed system was significantly related to major characteristics, such as the gender, age, and education of the population.

Since adolescent patients may not receive the treatments voluntarily, this study evaluated the communicative functions of our system for patients of different ages. Our experimental results revealed that the communication capabilities could have significantly different results. In the experiments, the patients were first divided into an adult group (ages greater than or equal to eighteen) and an adolescent group (ages of less than eighteen), and then a paired t-test was used to evaluate the four aspects discussed in the previous subsection. The same groups of units were tested twice to generate the results of the analysis.

Next, we discuss the results of the experiments. The *p*-values of the paired t-test all were lower than 0.05 with a 95% confidence level. That is, the four aspects were significantly different between traditional communication and the proposed system for the age classifications. Note that the final aspect (rate of change in intentions) had a slight difference between the adolescent and adult groups (the adolescent group had 0.015, and the adult group had 0) for traditional communication and the proposed system. It is believed that for the aspects of the rate of change in intentions, adults may have accepted orthodontic treatments more voluntarily than those in the adolescent group; that is, the adolescent group may not have had the intention to receive treatments because they were asked to receive them by their parents and elders, and were not doing so not of their own will. In sum, since the *p*-values were all lower than 0.05; these patients gave positive recognition to the system.

## 4. Discussion

Many medical information techniques have been introduced in healthcare. The technology of 3D simulation is a new trend in clinical systems that can provide patients and physicians with better communication before or during a medical process, especially for some irreversible, costly, or long-term treatments. With 3D simulation techniques, healthcare providers may be able to increase patients’ satisfaction, since they can evaluate their outcomes in advance and decide whether or not to accept a treatment. Another advantage is that 3D simulation can reduce the effort for communication and education between patients and physicians, resulting in healthcare providers having more spare time to engage in more activities related to the quality of care.

Generally, orthodontic treatments are complicated, and the entire course of treatment is not cheap or recoverable. The results of these medical treatments sometimes do not fully meet patients’ expectations, which can be due to information asymmetry between the patient and dentist, the recognition of the limits of treatment, and unreachable expectations from patients. This dissatisfaction often leads to medical litigation or even malpractice lawsuits. Thus, in order to increase the quality of treatment, reduce the gap between results and expectations, and minimize the generation of medical disputes in orthodontics, broad and effective communication between patients and physicians is very important.

This study proposes an efficient and effective 3D simulation system for orthodontics that can help dentists facilitate communication with patients to better understand treatments and their results. Comparing the literature with our study, the aim of the study [10] is to compare online and traditional learning methods in relation to orthodontic knowledge and skills, but no simulation system is proposed. Another study [14] is a clinical paper focusing on the orthodontic tooth movement (OTM), which is a highly regulated process that coordinates bone resorption. No computer-related materials are mentioned in that study. Our study adopts its orthodontic knowledge of tooth movements and the six forces used during orthodontic treatments. Some literature paid attention to dental simulation like ours. The study [13] surveyed the virtual reality used in dental education for teaching clinical skills in preclinical training for several subjects, such as endodontics, prosthodontics, periodontics, implantology, and dental surgery. Several virtual reality products were used in the above dental treatment procedures but they all do not mention the orthodontic treatment and neither propose a system for comparison with ours. Similarly, the study [11] is also a survey paper, stating that orthodontic treatment needs to manage a vast quantity of data, and computer science can be a big help in almost every aspect of the orthodontic practice, research, and education. However, it does not propose a system for orthodontics.

As we knew until now, there are two studies in the literature closely related to our study. The first is the study [15] that implements an interactive simulation system for training and treatment planning in orthodontics. That study focuses on training the orthodontists when they need to estimate the loading conditions that will achieve the desired tooth movement. It uses cephalometric measurements and dental cast data to validate its simulator and investigates the relationship between different mouth measurements, tooth movements, and appliances. The study focuses on the training of the orthodontics in the visual predictions of the final positions of the teeth under force loading in terms of clinical aspects; it does not focus on patient-physician communication like ours. Our study expresses the evolution of the orthodontic treatment step-by-step from the pre-treatment status, through several in-treatment statuses, to the post-treatment status for facilitating the communication. The second study [12] develops a 3D animation CAI (computer assisted instruction) system, which collects many past orthodontic cases and creates a 3D animation for each case. When encountering a new case, a most-similar past case transformed into 3D animation is selected based on the distance between the new and past cases. Then, this most similar case is shown for the patient-physician communication. Note that only the animation of the past cases is shown without any tuning. Our paper adopts the concepts of that paper but more flexible. In detail, our paper not only provides the past cases in 3D animation but also provides just-in-time tuning adapted to the current patient. Since the symptoms of patients may vary, tuning of the simulation model and rendering the results in time can provide a more accurate description in the treatment procedure, tooth arrangement, morphing of the gums with movement of the teeth, the placement of arch-wires and brackets, and changes of facial appearance, etc. It is believed our study has higher acceptance and satisfaction than that of this study.

There are some limitations in the system: (1) the resolution of all 3D models in this system is acceptable, but 3D models with a higher resolution are only feasible on a larger and fast computer; (2) this system referenced the literature and orthodontic textbooks to summarize and display 14 symptoms belonging to Angle class II. Some rare symptoms may not have been included; (3) the relationship between social class and the uptake of orthodontic treatment was not investigated in the questionnaire. In fact, the working class people receive inferior and less dental care than middle class people. In most cases, the socioeconomic status dominates whether to accept the high-cost dental service rather than the patient–dentist communication system, especially if the payment is not covered by the insurance. In future work, new 3D models can be added, such as mandibles, maxilla, and the whole face, in order to realistically emulate parts of the head and soft tissues and provide more comprehensive and realistic simulations.

## 5. Conclusions

This study proposed a new framework for a 3D simulation system for orthodontics. The system can simulate the profile of a face and the arrangement of teeth, gums, arch-wires, and brackets to emulate the current status of a clinical patient, and it can simulate and display the procedure of the change from malocclusion to normal occlusion and the pre-treatment, mid-treatment, and post-treatment status. By using these simulation functions, dentists can vividly communicate with patients, and patients can better understand the processes and results of treatments. In addition, through the intuitive operation provided by the system, it provides just-in-time simulation results. The results of the evaluation of the system showed that it received positive recognition from dentists. Clearly, the system is an effective tool for communication between dentists and patients and reduces the gap between the expectations of the patients and orthodontic limitations, and it increases the satisfaction of patients.

## Figures and Tables

**Figure 1 healthcare-11-01433-f001:**
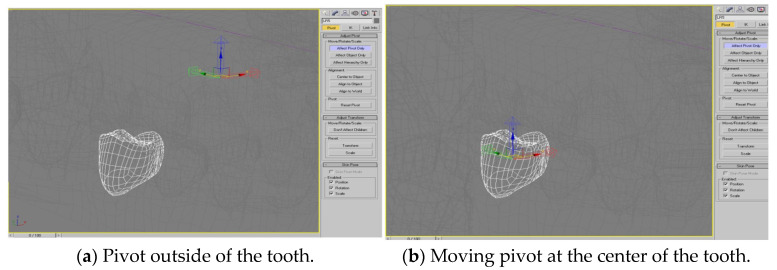
Moving the pivot from the outside of a tooth to the center of mass of the tooth.

**Figure 2 healthcare-11-01433-f002:**
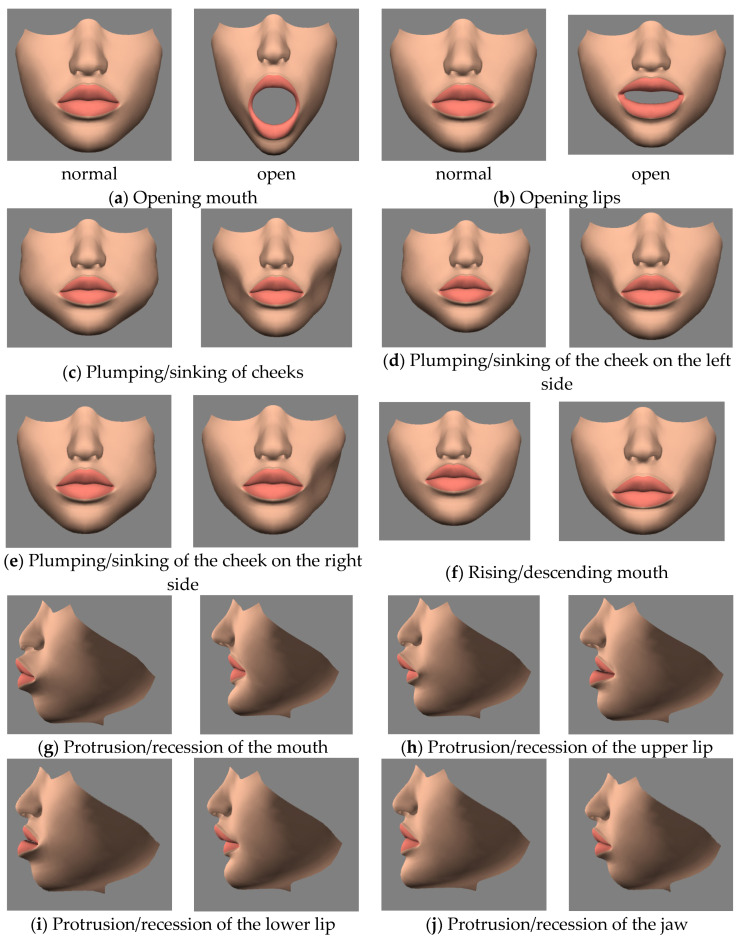
The morphing results for the face used in the system. Figure (**a**–**j**) are 10 different morphing results for the mouth.

**Figure 3 healthcare-11-01433-f003:**
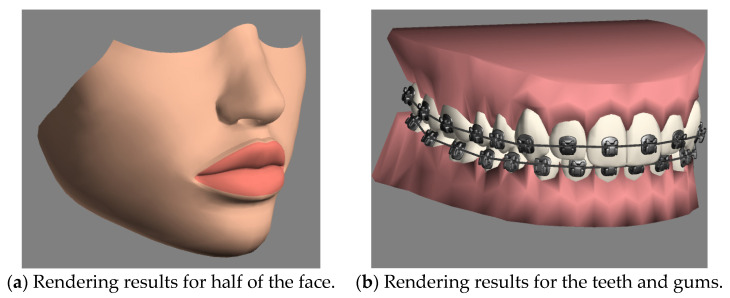
The rendered results after the editing steps on the basic 3D models for the lower half of the face and the mouth containing teeth, gums, arch-wires, and brackets.

**Figure 4 healthcare-11-01433-f004:**
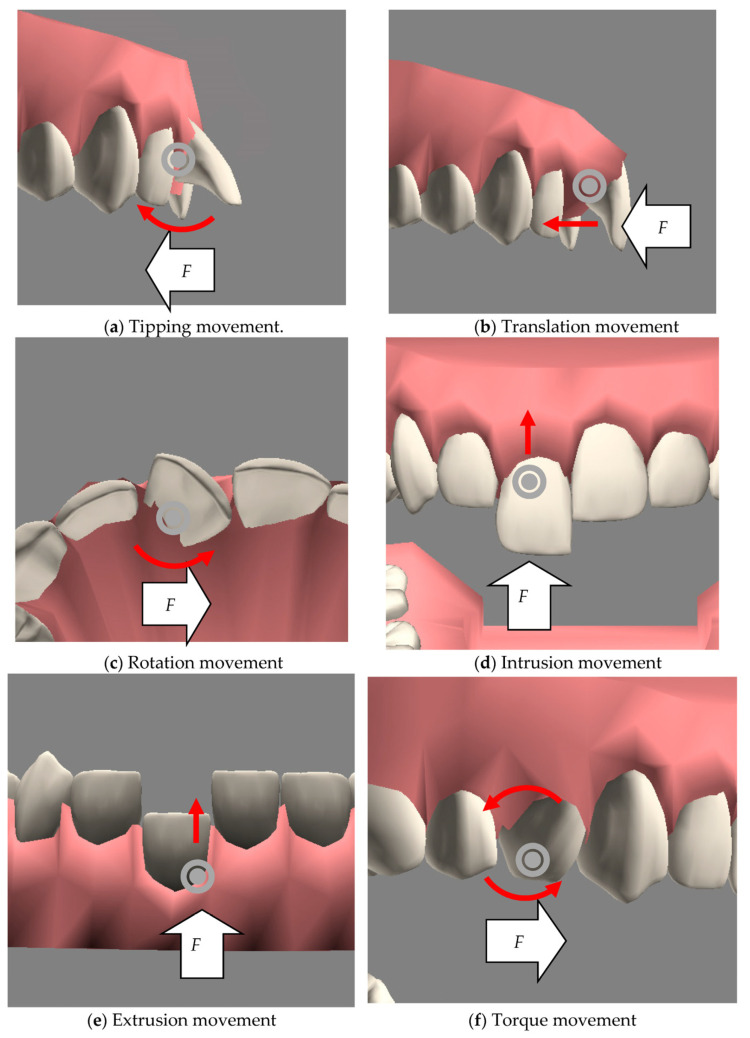
The six types of tooth movements.

**Figure 5 healthcare-11-01433-f005:**
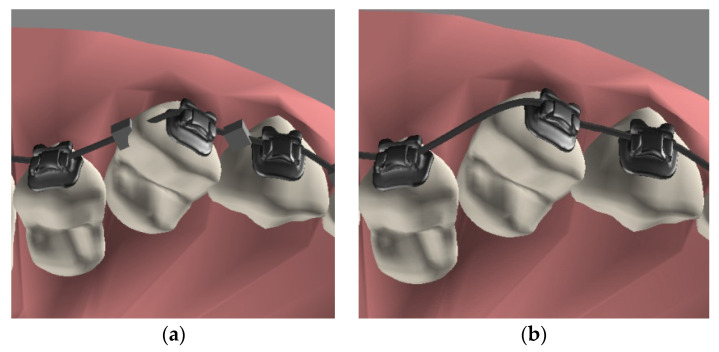
A model of an arch-wire with controlling nodes. (**a**) The arch-wire penetrated the tooth before the controlling node was adjusted to the correct position. (**b**) After hiding the controlling node, the arch-wire model looked like that in a real case.

**Figure 6 healthcare-11-01433-f006:**
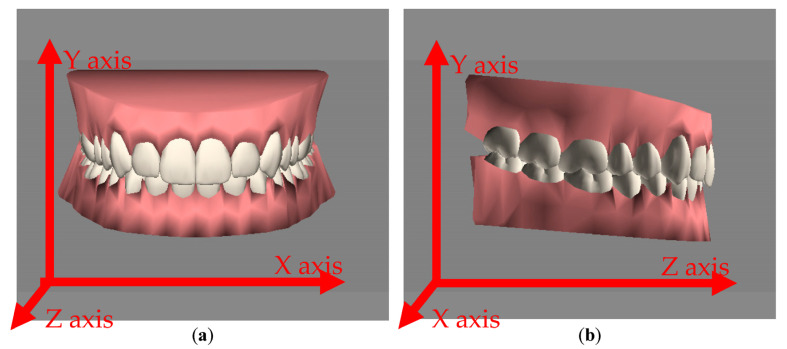
The *x*, *y*, and *z* axes. (**a**) Front view with the *x* and y axes. (**b**) Right view with the *y* and *z* axes.

**Figure 7 healthcare-11-01433-f007:**
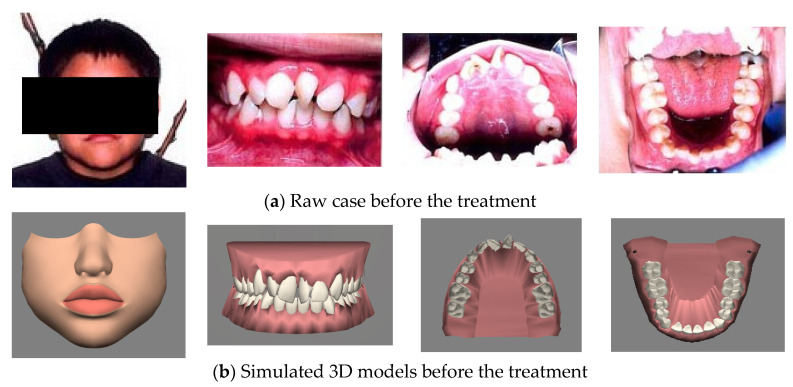
Simulated 3D model and their corresponding real case in pre-treatment.

**Figure 8 healthcare-11-01433-f008:**
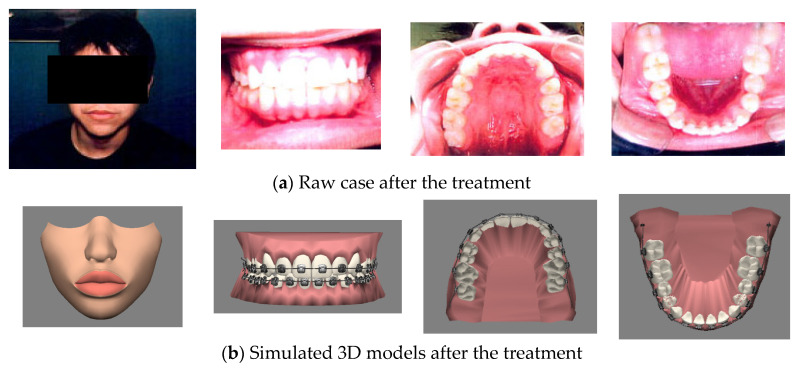
Simulated 3D model and their corresponding real case in in-treatment.

**Figure 9 healthcare-11-01433-f009:**
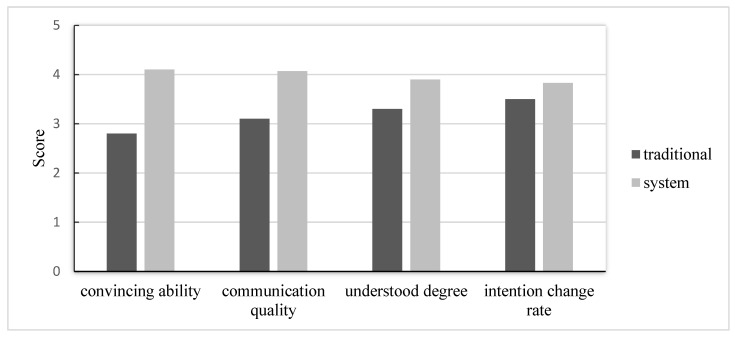
Results for the four aspects in the two phases of questionnaires.

**Table 1 healthcare-11-01433-t001:** Each of the 3D models with different numbers of polygons and vertices.

Name of 3D Model Part	Number of Polygons	Number of Vertices
All brackets and arch-wires	140,928	70,848
32 (or 28) teeth	30,296 (or 26,376)	15,212 (or 13,244)
Lower half of the face	3320	1732
All gums	1060	1804

## Data Availability

Not applicable.

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
