# Peer review of "Effective Patient–Dentist Communication with a Simulation System for Orthodontics"

_healthcare, 2023, doi:10.3390/healthcare11101433_

Round 1
Reviewer 2 Report
In general, this is a poor written and disorganized manuscript. The format of this article needs to divided 4 sections: introduction, materials & methods, results, and discussion.
Without the result section in the text, this article need to provide the scientific evidences of this 3D dentist – patient communication system for simulation of orthodontics.
In addition, there is also no any information of questionnaire such as reliability, validity, and even results.
Without the solid evidences, therefore, this manuscript can not be recommended for the publication in Healthcare.
Reviewer 3 Report
Thank you for this helpful contribution. I applaud the effort of promoting studies for the investigation of Patient - Dentist communication. I appreciate their methods including study design and data analysis. I write some comments below that could benefit the article.
Discussion??? It is a main section. it is very important to compare the results with similar studies.
Results. Information is shown in duplicate. If it appears in the figures and tables, delete the information from the text.
References. It is advisable to reference articles published in scientific journals. The section is very short.
Congratulation again!!!
Round 2
Reviewer 1 Report
Many thanks for your resubmission. You have put a lot of effort into it and the article is now organised in a much more understandable and comprehensible way. My questions have been answered to my satisfaction. Best regards
Author Response
Thanks for your precisous comments, and it is our mission to let the paper be understandable and comprehensible. We invite the English editing service to promote its readability. We hope we have the chance to share the study to the academic through this prestigous journal. Thanks a lot.